# Multi-Criteria Decision Analysis for Nautical Anchorage Selection

**Danijel Pušić \* and Zvonimir Lušić** 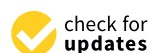

Faculty of Maritime Studies, University of Split, 21000 Split, Croatia; zvonimir.lusic@pfst.hr
\* Correspondence: danijel.pusic@pfst.hr

**Abstract:** Considering that moorings and anchorages for vessels have recently become an important factor in nautical tourism, the selection of their locations is a complex and demanding process. This paper examines numerous criteria from different perspectives to determine the most favourable/optimal locations for nautical anchorages, meeting the conditions and recommendations of professionals from several domains, by applying the methods of multi-criteria analysis. The goal of solving the problem this way is to meet the expectations of future users, spatial planners, possible investors, and concessionaires interested in doing business in these areas, as well as entities that strive to preserve and protect marine and underwater animal life and the environment by preventing their degradation and pollution. However, since there are no precisely defined recommendations for the establishment of nautical anchorages, in the procedures for determining the locations of nautical anchorages, it is possible to use general criteria they must fulfil. The best locations for nautical anchorages may be found, and this research represents a transparent, repeatable, and well-documented approach for methodically solving the problem. This is demonstrated by a comparison of many methods of multi-criteria analysis, utilizing a variety of parameters. On the other side, this calls for proficiency in a wide range of disciplines, including architecture, geodesy, marine safety and transport, architecture, biology, ecology, mathematical programming, operational research, information technology, environmental protection, and others. The best locations for nautical anchorages should be chosen based on the size and number of vessels, available space, depth, distance from the coast, level of protection of the anchorage waters, and many other limiting factors, keeping in mind that the spots which simultaneously satisfy a greater number of significant criteria are preferable. Using multi-criteria analysis methods (AHP (Analytical Hierarchy Process) and TOPSIS (The Technique for Order of Preference by Similarity to Ideal Solution)), evaluating and classifying criteria as well as assigning weight values to selected criteria, this paper investigates the possibility of obtaining the best locations from a group of possible ones. The most important factor when applying multi-criteria analysis methods refer to the following: vessel safety (navigation), hydrometeorological, spatial, economic, and environmental criteria. The main contribution of the paper displays in the proposal to optimize the decision-making process, when determining the optimal locations of nautical anchorages, in accordance with previously defined criteria.

**Keywords:** nautical anchorage; multi-criteria analysis; criteria; optimization of spatial locations; weight coefficients; concession fields; AHP; TOPSIS

## 1. Introduction

The construction of a new nautical anchorages system is a large and long-term investment, and therefore, the determination of new locations is a critical point on the way to the success or failure of the system by exploiting them. One of the main goals when considering the locations for nautical anchorages is to find the most suitable location which can increase the desired conditions defined by the selection criteria.

After finding the nautical anchorages locations, an attempt is made to optimize the number of objectives in determining the suitability of a particular location for a defined

system. Such optimization often involves many and sometimes even contradictory factors. Some of the important factors increase the complexity of choosing the right/best among the many possible locations.

The process of selecting the location of nautical anchorages involves a number of perplexing key factors that include maritime, spatial, hydrometeorological, traffic technical, economic, social and other issues, such as sea, coastal, environmental protection, etc. Due to the complexity of the process itself, the simultaneous use of several tools is required for decision support, such as Expert Systems (ES), Geographic Information Systems (GIS), and Multi-Criteria Decision-Making methods (MCDM).

In order to obtain a safe and comfortable stay while anchoring or using certain specific facilities at sea, sailors (both professionals and amateurs) need to take into account many location aspects, especially those considering protection from wind, waves, sea currents, etc.

On the other hand, from the point of view of planning and space utilization, planners and management of local and regional communities, who want to optimize the space at sea, should have in mind a number of factors that enable them to plan the space in the best way, observing a whole series of other factors and especially those who take into account the protection of nature, the sea and coastal areas, the environment, traffic and technical conditions, the current situation on the ground, etc.

Therefore, an attempt is made to compromise on the wishes of future users of nautical anchorages (professional sailors, amateurs and future concessionaires of nautical anchorages, spatial planners, administrations of local communities, and others) by creating a very sophisticated system of interconnections and interdependence, with the aim of selecting the best spaces in the water area predetermined for anchoring vessels while taking advantage of all the benefits that these locations can provide in order to fulfil all expectations or most of them, without disturbing surrounding space, sea, coast, and environment. The former represents the research problem of this work.

The previously defined problem can be solved by applying MCDM methods in order to optimize the selection process of the best locations for setting up nautical anchorages planned to be used as concession fields in the area of Split-Dalmatia County (Croatia).

Given that previous research conducted both in Croatia and around the world shows that there is no unified methodology for selecting the best locations for nautical anchorages, neither for their layout nor from the navigational and safety point of view, for vessels and both their crew and passengers there should be a social and a scientific contribution.

The paper will also present an overview of the factors affecting the anchorage area and prove that the correct selection of the location and the construction of the necessary facilities remarkably affect the safety of people, vessels, and the marine environment. By applying multi-criteria methods and analysis, the best 15 locations of nautical anchorages will be selected from a set of possible 86 available.

The proposal for a systematic solution to the problem posed in this paper was accomplished by applying two methods of multi-criteria analysis 1. AHP (Analytical Hierarchy Process) and 2. TOPSIS (The Technique for Order of Preference by Similarity to Ideal Solution). Using the mentioned methods, several conflicting criteria are taken into account, the weighting values and mutual dependence of which were previously determined on the basis of a survey of future users.

The paper is structured as follows: The motive for the research and the existing problems are described in the introduction, while the second chapter lists materials, methods, previous research, and the most important recommendations for determining the locations of nautical anchorages. The basic steps of the multi-criteria decision-making methods used in this paper are defined in the third chapter, while the fourth chapter analyses the case study. The fifth chapter indicates the most important validation, testing, comparison, and result analysis, while the sixth presents a discussion of the entire procedure.



## 2. Materials and Methods

### 2.1. Methodology and Research Plan

Research implementation includes research plan overview and investigation of previous research. The process of collecting and processing data in the field was conducted simultaneously: (a) by gathering the opinions of future users of nautical anchorages through survey research and (b) through the collection, analysis, processing, and storage of data by the author/s at the locations of nautical anchorages for the area of Split-Dalmatia County (in the period from 2018 to 2022). Criteria were selected based on both obtained average scores that the respondents (users) assigned to certain aspects of nautical anchorages and available data on nautical anchorages previously collected (by the author/s). Furthermore, they have defined the goal of each of them as well as the weight values of each criterion and their mutual relations. Both different MCDM methods (AHP, TOPSIS) were applied.

Validation, testing, comparison, and result analysis are shown in the chapter Results. The research methodology in this paper consists of two phases.

In the first phase, the attitudes and opinions of the nautical anchorages users were examined on the basis of a survey questionnaire. The survey questionnaire was made available to visitors of nautical tourism ports and nautical anchorages in Split-Dalmatia County and the entire international community and was published on the pages of the International Association of Maritime Universities [1].

Opinions were collected by forming a questionnaire of future users of nautical anchorages who evaluated five groups of elements (and a total of 18 sub-elements), namely the following: safety (navigational—6 sub-elements); hydrometeorological (with 2 sub-elements); spatial (with 4 sub-elements); economic (with 2 sub-elements) and environmental (ecological) (with 4 sub-elements), which were electronically sent to respondents in the time period from November 2022 to January 2023.

There are a number of factors that need to be considered when identifying suitable sites for anchoring vessels and will often require consultation with a wide range of stakeholders. The most important guidelines and recommendations on factors to consider [2,3].

The survey questionnaire was created and distributed via the web ArcGis survey 123 [4] service. Answers to the questionnaire were received from 74 respondents.

In the second phase, after arranging the data (as a result of survey research) and assessing the significance of certain factors from the perspective of future users of nautical anchorages, the most important criteria were selected and weighted values assigned to each of them.

Then, two different methods of the MCDM method were applied to the input data of 56 bays with 86 locations, i.e., fields of nautical anchorages (variants), for each of which 17 data were known. Essentially, the aim is to select the best locations for nautical anchorages out of the 86 taken into consideration, respecting the criteria that are the most significant and have the most influence. Both methods of the MCDM were implemented using the programming language R (version 4.2.2) [5] and heuristics.

The most important elements on the basis of which the criteria were defined and grouped into five groups are shown in Figure 1. The criteria were derived from various sources, including the Queensland Government's Anchorage Area Design and Management Guideline [3] and The World Association for Waterborne Transport Infrastructure (PIANC) [6]. Later, they were grouped into five categories based on their nature and impact on the selection process.

The parameters for the multi-criteria analysis were obtained from the factor criteria. The most important factors, i.e., criteria, are the following: 1. the surface of the field; 2. the surface of the bay; 3. the percentage share of the field surface in the bay surface; 4. the degree of protection (from wind and waves) of the bay; 5. the distance from the coast; 6. the number of anchorage fields in the same bay; 7. the existence of maritime traffic; 8. official anchorages; 9. the existence of underwater cables and pipelines; 10. the risk of collision; 11. depth; 12. the level of sea changes and the existence of sea currents; 13. proximity to public ports; 14. proximity to existing berths; 15. environmental elements (Environmental

Network Natura 2000); ref. [7] 16. damage from anchoring the vessel to the seabed; and 17. archaeological sites.

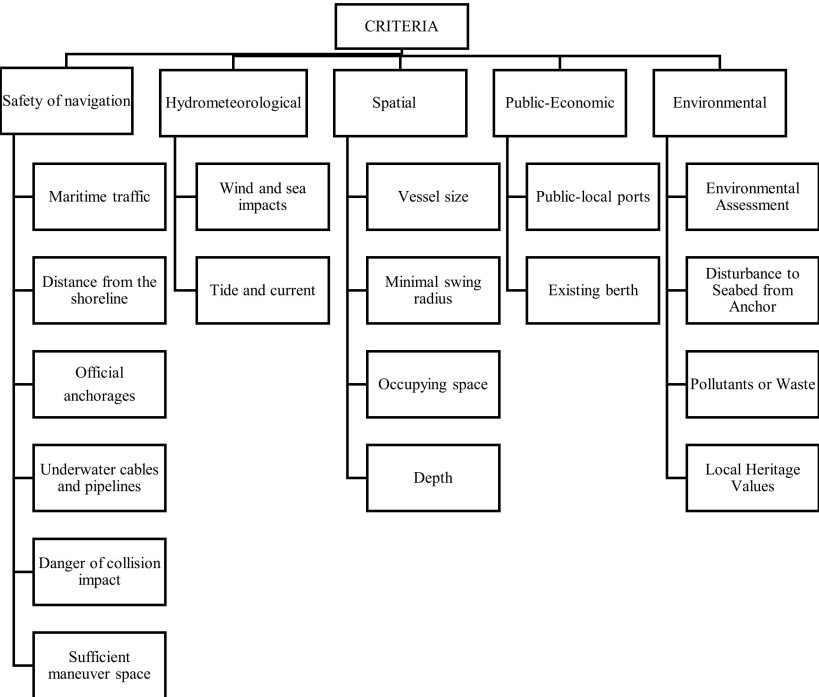

**Figure 1.** The most important criteria of nautical anchorages [6].

The analysis is limited and carried out according to a limited number of criteria (Figure 1) and certain weight values (based on determining their importance and impact). The selected criteria are unique and coherent, although some are related and interdependent. Some criteria have a greater impact and importance, so they have been singled out. This is the case for, e.g., criteria determined by a border (distance from the coast or depth of the sea, etc.), some of which are imposed by legal regulations, etc.

### 2.2. Background

The proposed scientific literature for the determination of the best locations for nautical anchorages based on MCDM is relatively small or does not exist at all. There are, however, studies related to methods of MCDM for areas of spatial planning, for example, nautical tourism ports or methods for processing spatially distributed data. Regarding the creation of this paper represents a special challenge to the authors, especially in terms of applying the following: multiple methods of MCDM in parallel over; more numerous input data; using a multi-criteria analysis methodology that combines several different criteria (safety, hydrometeorological, spatial, economic, and ecological); considering the selection of the best locations of nautical anchorages both from the point of view of users and from the point of view of concessionaires, i.e., future investors; contributing to the expansion of scientific methodology related to the subject of research.

In the doctoral dissertation [8], the criteria for assessing the possible impacts of the functioning and operation of the liquefied gas terminal on the marine environment are defined. Using the MCDM, a model that monitors the effect of the selection of individual criteria on the marine environment was created. The pre-elimination criteria were determined in order to reduce, i.e., limit, the observed space from which representative locations previously evaluated by certain criteria were selected. The observed problems were solved with the following methods: GIS, MCDM, and ES, using mathematical models and a regional model of the complete maritime system as a tool.

The research carried out in the doctoral dissertation [9] is related to finding an optimal model with five scenarios and undertaking certain measures for the development of county



and local ports in relation to the complementarity of the spatial concept of the port and the city. For the purposes of defining and setting up the model, the most important indicators and measures that affect the level of spatial planning of the port and the city were used. The success of the set model was tested and demonstrated using the port of Rovinj as an example.

The paper [10] presents a conceptual framework for the inclusion of multiple criteria in the assessment of a dry port for developing countries from the perspective of multiple stakeholders. The framework of the work is presented in four steps covered by preliminary research, namely the following: 1. stakeholders are grouped into groups; 2. sub-criteria related to the location of the dry port are listed; 3. individual criteria and sub-criteria are explained; and 4. an MCDM analysis is carried out.

The paper [11] investigates the logistical possibilities of offshore wind farms, specifically the physical characteristics, connections, and appearance of the port to support the phases of installation, operation, and maintenance of offshore wind farm projects. The relative importance of these criteria was determined using the AHP method. The AHP methodology is then applied in a case study as a decision-making tool to enable decision-makers to assess the suitability of a range of ports for offshore wind farms in the UK North Sea.

The paper [12] presents a multi-criteria spatial evaluation (SMCE—Spatial Multi-Decisional Evaluation) intended to identify suitable areas at the regional level for setting up middle-sized coastal fish farms in the Ligurian Sea. The SMCE process follows an integrated approach that can potentially be adapted and applied to any coastal system. The selection of the location is based on the definition of criteria that assess their suitability and how their conditions relate to the entire research area. The results show that SMCE, and especially the procedure, enables the identification of the most suitable areas, which solves the complicated problem of spatial selection of an appropriate location in a simple, fast, and efficient way.

The study [13] is an expert basis for amendments and additions, i.e., the adoption of a new Spatial Plan of the Split-Dalmatia County based on navigational and meteorological features, as well as technical-technological and traffic-navigation features, maritime safety measures, the Natura 2000 Habitats Directive [14], the Register of Strictly Protected Species, technical-technological types of anchorages and the organization of anchorages using expert analysis, with possible future locations defined as nautical anchorages. The cited work also describes the prerequisites that the investor must fulfil in order to obtain the necessary permits and papers to start work.

The research project [15] serves as an expert basis and support for the authorities of the Split-Dalmatia County and the Public Institute for Urbanism in planning the future development and location of ports and anchorages for nautical tourism. The cited project considers the development possibilities of the port of nautical tourism in the Split-Dalmatia County from the point of view of optimal use and environmental protection. All elements of supply and demand of nautical tourism were determined by SWOT (S-Strengths, W-Weaknesses, O-Opportunities, T-Threats) analysis. By defining the advantages, disadvantages, strengths, and weaknesses of Croatian nautical tourism, it is possible to determine its strategy and direction of development. The application of MCDM methods of ports of nautical tourism in the Split-Dalmatia County included a systematic study of proposed micro locations, which resulted in the selection of land and island locations. The next step was the definition of criteria and sub-criteria for their acceptability, and finally, a qualitative and quantitative evaluation of each proposed location was performed.

Report [16] offers resolving a location selection problem by means of an integrated AHP-RAFSI (Ranking of Alternatives through Functional mapping of criterion sub-intervals into a Single Interval) approach. The main goal of this report is to find the optimal EMS (emergency services) as the locations that provide the least response time. The methods used vary and mainly relate to the decision-making variables taken into account. Multi-criteria decision making (MCDM) is used in the case of emergency centre allocation in

Libya. MCDM approach was implemented in two steps. AHP was adopted in the first step to determine the criteria weights, while the results of AHP showed that the response time had the highest weight among other criteria. A ranking of different alternatives was conducted in the second step using RAFSI model to choose the optimal location. Model ranking clearly indicated road-network as the best alternative to locate EMS centres.

The aim of research [17] is to select the best supplier in LISCO in Libya using the Rough AHP method. With increasing awareness of sustainability aspects, both from the academic sector and from the industrial sector, the authors believe that sustainable decision-making techniques for selecting the best suppliers should improve supply chain competencies and help companies maintain a strategically competitive position. The results showed that the Rough AHP method is capable of improving the quality of decision-making by making the process more rational, explicit, and efficient and that in future work this effective method can be generalized to other companies throughout Libya to facilitate the measurement of sustainability performance, consequently providing the company with a robust system while maintaining ecological sustainability.

Research of scientific and professional literature and data that thematically deals with nautical anchorages, criteria, and selection of new locations, especially in Croatia, indicate that there is a modest amount of officially precise collected data on their condition and registered physical traffic in organized nautical anchorages, especially those in Split-Dalmatian counties. In addition, the level of quality of the services provided in them is low, so it is necessary to use more modern analytical decision-making methods to collect accurate and precise data in order to, on the basis of the data thus obtained, use the methods and tools of multi-criteria analysis in the optimal and systematic way of choosing the best locations of nautical anchorages.

Previous research conducted in the world and Croatia on the topic of research problems (determining the best locations for nautical anchorages) showed that there is no single methodology for selecting the best locations for nautical anchorages, as well as their arrangement, especially regarding the safety aspect. Most of the research focuses only on economic factors when choosing anchorages, including other aspects such as tourism, legal, social, sustainable development, protection of the marine environment, seas, and coasts, mostly ignoring safety factors when choosing criteria. The anchorage site selection can be seen in the analysis of the fundamental concept of navigation safety as an important and indispensable part of spatial planning.

Unlike previous research, this work systematically and uniquely takes into account safety (navigation), but also hydrometeorological, spatial, economic, ecological, and other conditions and criteria in the process of selecting the best locations for nautical anchorages. At the same time, the greatest importance is attached to the fundamental concept of safety of navigation and staying at anchorages, as one of the most important factors of the purpose of anchorages and an important element of spatial planning.

### 2.3. Recommendations for Setting up Nautical Anchorages

In order to determine the optimal solution for the selection of the best locations in the Split-Dalmatia County and the adequate application of MCDM methods, the most important recommendations [18] can be summarized in the list below:

- It is necessary to take into account the existing situation, position, and size of the fields and respect as much as possible the existing boundaries of concession fields, initiatives, conceptual solutions, and proposals if they do not contradict the general principles of maritime safety and protection of the marine environment;
- Anchorages must not interfere with maritime traffic, i.e., both existing and future routes of transit and terminal waterways, and generally need to be in accordance with spatial plans;
- The boundaries of the anchorage must be at a safe distance from the shore, at least twice the width of the largest ship that is expected to sail from a safe isobath, and need to take into account other factors such as possible traffic of other ships, boats and/or

beaches, bathing areas, and other facilities expected between the anchorage and the shore, where bathers, swimmers, divers and other persons in the sea could endanger the safety of the anchorage;

- The anchorage must be in an area of sufficient depth, whereby the depth of the lowest low water level of living sea minnows must not be less than the expected draft of the ship increased by 1 m;
- The surface of the anchorage may not occupy more than 50% of the bay (exceptionally up to 75%) if it is a bay where there are no other moorings, beaches, infrastructure, or other facilities or activities that require access from the sea or land;
- In the water area, from the nautical anchorage towards the coast and at a distance of up to 150 m from the nautical anchorage in the direction of the open sea, there should be no other artificial installations or structures, including anchorages or measures to direct maritime traffic (regulations on the safety of maritime navigation in internal sea waters and the territorial sea of the Republic of Croatia) [19], and the manner and conditions of supervision and management of maritime traffic;
- Anchorages may not be in the area of official anchorages on the map designated by the port authorities/MMPI (Ministry of the Sea, Transport, and Infrastructure of Croatia) [20] and in places of shelter if they are determined by the ordinance on places of shelter [21];
- Avoid anchoring in the immediate vicinity of submarine cables, submarine installations, and other places where anchoring is prohibited;
- Avoid positioning the anchorage that would be a potential danger of collision, impact, injury, and other risks;
- Anchorages must not limit the manoeuvring space for ships outside the anchorage that need to manoeuvre, and should also provide sufficient manoeuvring space for ships arriving or departing from the anchorage;
- Anchorages must not interfere with the existing moorings of the local population.

## 3. Methods of Multi-Criteria Analysis When Determining the Optimal Position of Anchorage

### 3.1. Definition and Description of Multi-Criteria Decision-Making Methods

Multi-criteria analysis plays an important role in the selection of the best variants for finding the best areas (locations) in many areas of spatial planning, optimization of urban and non-urban structures, etc.

According to the manual on multi-criteria analysis (multi-criteria analysis: a manual) published by Communities and Local Government London, the method of MCDM is defined as an approach that explicitly shows all options and their contributions, on the basis of which assistance in the decision-making process is subsequently realized [22,23].

Multi-attribute decision models persist in determining the optimal variant from a set of finite variants $V = \{V_1, V_2, \ldots, V_m\}$ which are compared with each other with respect to assigned numerical or non-numerical values belonging to the finite set of criteria $C = \{C_1, C_2, \ldots, C_n\}$. Each criterion can aim to reach a maximum or minimum value.

For decision problems with multiple attributes, in which the matrix of consequences contains heterogeneous data, numerical or non-numerical, the homogenization of these data is performed by the normalization process [24], which transforms the matrix of consequences into the matrix $R = (r_{ij})$ $i = 1, m; j = 1, n$, as well as into elements on a certain interval, for example from 0 to 1 $[0, 1]$ or from $-1$ until $+1$ $[-1, +1]$, etc.

In almost all MCDM problems, there is information about the level/degree of importance of each criterion expressed by the vector $P = \{p_1, p_2, \ldots, p_n\}$ that represents the assessment determined by the decision-maker for each criterion.

Any multi-attribute decision problem can be expressed by a matrix $A$, which is called a consequence matrix (decision matrix) (Table 1), with elements $a_{ij}$ showing the evaluation (consequence) of variant $i$, $i = 1, 2, \ldots, m$ $(V_i)$, by criterion $j$, $j = 1, 2, \ldots, n$, $(C_j)$.

**Table 1.** Decision or consequences matrix.

| $V_j$ $\diagdown$ $C_i$ | $C_1$ | $C_2$ | $\ldots$ | $C_n$ |
|---|---|---|---|---|
| $V_1$ | $a_{11}$ | $a_{12}$ | $\ldots$ | $a_{1n}$ |
| $V_2$ | $a_{21}$ | $a_{22}$ | $\ldots$ | $a_{2n}$ |
| $\ldots$ | $\ldots$ | $\ldots$ | $\ldots$ | $\ldots$ |
| $V_m$ | $a_{m1}$ | $a_{m1}$ | $\ldots$ | $a_{mn}$ |
| $P$ | $p_1$ | $\ldots$ | $p_n$ | |

MCDM methods can be classified into three categories [25], namely:

- Direct methods;
- Indirect methods;
- Methods that use a distance for the construction of hierarchies.

Direct methods build a function defined on a group of variants with real values and select the variants for which the objective function f has the biggest value.

Indirect methods determine a hierarchy on a set of variants based on an algorithm. Methods that use distance choose the variant that is closest to the ideal solution.

In this paper, AHP (direct method) and distance methods (TOPSIS) are used.

*3.2. AHP*

AHP is an MCDM method originally developed by Prof. Thomas L. Saaty in the 1970s and has been extensively studied and refined ever since, representing one of the most popular analytical techniques and providing a comprehensive and rational framework for structuring and solving multi-criteria decision problem [26].

According to [27], the steps of the AHP method that are to be followed during implementation are described below:

Step 1: Develop a decision hierarchy by decomposing the entire problem into a hierarchy of parameters or criteria;

Step 2: Prioritize among the parameters or criteria of the hierarchy by making a series of judgments based on pairwise comparisons. In this step, the preferences among the criteria are evaluated based on Saaty's scale [27] from 1 to 9, and from 1/9 until 1.

Step 3: Synthesizing the judgment to obtain a set of general priorities for the hierarchy. In this step, the weighted results of the criteria are calculated, which give a relative ranking of the parameters or criteria;

Step 4: Comparing qualitative and quantitative information using informed judgments to derive weights and priorities to check consistency of judgments;

Step 5: Selecting the best alternative based on the available sample data and calculating the final score of each alternative.

In the MCDM method AHP, the decision-making problem is hierarchically structured (Figure 2), given that the decision-making problem is decomposed into subproblems that are analysed independently. At a certain level of the hierarchy, each element (criterion or alternative) is compared with other elements of the same level.

Therefore, based on the matrix of real values (estimates) determined by $x_{ij}$ values, for each criterion ($C_j$, j = 1, *n* where n represents the total number of criteria) and each alternative A ($A_i$, i = 1, *m*, where m represents the total number of alternatives) of the decision-making relationship, the input data are represented by the decision-making matrix *D* shown mathematically (Equation (1)).

$$C_1 \quad \ldots \quad C_n \quad D = \begin{matrix} A_1 \\ \ldots \\ A_m \end{matrix} \begin{pmatrix} x_{11} & \ldots & x_{1n} \\ \ldots & \ldots & \ldots \\ x_{m1} & \ldots & x_{mn} \end{pmatrix} \tag{1}$$

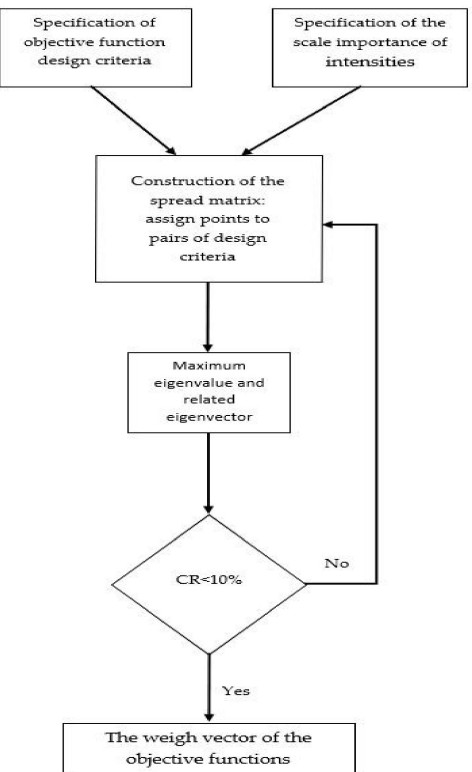

**Figure 2.** AHP flow diagram [28].

The formulas used during solving the matrices in the AHP steps are given below, as well as the flowchart of the AHP steps [29].

The results of the comparison by pairs of criteria are presented by a square matrix of comparison A of order $n \times n$, where n is the number of observed criteria (alternatives at a later stage). The matrix element $a_{ij}$ of matrix $A$ represents the relative importance of criterion i in relation to criterion $j$. If $a_{ij} > 1$, criterion i is more important than criterion $j$, while the reverse is true for $a_{ij} < 1$. If two criteria are of equal importance, then $a_{ij} = 1$.

For consistency, $a_{ij} = 1/aij$ holds for each $i$, $j$.

Therefore, $a_{ij} = 1$ holds for every $i$. Additionally, due to transitivity, $a_{ij} = \alpha_{ik} . \alpha_{kj}$ should hold for every $i$, $j$, $k$. The preferences or relative importance of decision-makers are expressed by Saaty's scale of relative importance, i.e., numbers from 1 to 9, and from 1/9 to 1.

Based on Saatya's scale of relationships between criteria, matrix $A$ was formed. (Equation (2))

$$A = \begin{pmatrix} 1 & a_{12} & \ldots & a_{1n} \\ a_{21} & 1 & & a_{2n} \\ \ldots & \ldots & \ldots & \ldots \\ a_{n1} & a_{n2} & \ldots & 1 \end{pmatrix} = \begin{pmatrix} 1 & a_{12} & \ldots & 1/a_{n1} \\ 1/a_{12} & 1 & & a_{2n} \\ \ldots & \ldots & \ldots & \ldots \\ 1/a_{n1} & 1/a_{n2} & \ldots & 1 \end{pmatrix} \tag{2}$$

whose values along the main diagonal have values of 1.

The vector $B$ (Equation (3)) represents the sums of the elements of the matrix by rows, dimensions $n$.

$$B = \sum_{i=1}^{n} a_{ij} = b_i, j = 1 \tag{3}$$

Dividing each element of the matrix (of a certain column) with the elements of the vector $b$, the values of the normalized matrix $G$ are defined by (Equation (4)).

$$G = \begin{pmatrix} g_{11} & g_{12} & \cdots & g_{1n} \\ g_{21} & g_{22} & & g_{2n} \\ \cdots & \cdots & \cdots & \cdots \\ g_{n1} & g_{n2} & \cdots & g_{nn} \end{pmatrix} \tag{4}$$

By calculating the mean values of the elements of the normalized matrix for all columns of the same row, the elements of the vector of weighting coefficients $w$ of dimension $n$ are obtained.

The mean value of all elements of the normalized matrix by column represents a vector of weight coefficients, the sum of which is 1 (Equation (5)).

$$W = \begin{pmatrix} w_1 \\ w_2 \\ \cdots \\ w_n \end{pmatrix} = \sum_{i=1}^{n} w_i = 1, i = 1, n \tag{5}$$

The sum of the products of vectors $W$ and $B$ gives the value $\lambda_{max}$ which represents the maximum eigenvalue $\lambda \sum_{i=1}^{n} w_i \cdot b_{imax}$.

$A\omega = \lambda_{max}\omega$, forms the matrix of preferences, $\omega$ is the eigenvector of order $n$ representing the vector of weight values, while $\lambda_{max}$ represents the maximum eigenvalue.

The consistency index is calculated according to Equation (6).

$$CI = \frac{\lambda_{max} - n}{n - 1} \tag{6}$$

where $n$ is the number of parameters (criterion).

The consistency ratio is calculated according to Equation (7).

$$CR = \frac{CI}{CRI} \tag{7}$$

where *CRI* is the consistency index conditioned by the number of criteria.

When the consistency ratio is less than 10%, it is considered that the relationship between the criteria is consistent, and it is passed to the second stage of the AHP method (Table 2).

**Table 2.** Consistency ratio for the defined number of criteria.

| Number of Criteria | 1 | 2 | 3 | 4 | 5 | 6 | 7 | 8 | 9 | 10 |
|---|---|---|---|---|---|---|---|---|---|---|
| CRI | 0.00 | 0.00 | 0.58 | 0.90 | 1.12 | 1.24 | 1.32 | 1.41 | 1.45 | 1.49 |

In the process of selecting the best locations in the case study, the AHP method was applied. AHP implies that after the calculation of the consistency among the criteria, the normalization of the elements, the calculation of the mean value of the product of the vectors of normalized values of each variant (86 of them—$V_{ij}$, $j = 1, k$), the vector of weight values ($w_j$, $j = 1, k$), and a vector of final values of $v_i$ are defined by (Equation (8)).

$$v_i = \frac{\prod_{j=1}^{k} V_{ij} w_j}{k} \tag{8}$$

A higher value of the variant ($v_i$) determines a greater influence of the variant in the total rank of the variants (location).

*3.3. TOPSIS*

The TOPSIS method was developed by Hwang and Yoon in 1981 [30]. It is based on the idea that the chosen alternative should be the one with the shortest Euclidean distance from the ideal solution and the one with the greatest distance from the negative ideal solution. An ideal solution is a hypothetical solution for which all attribute values correspond to the maximum values in the data group that contain satisfactory solutions. A negative ideal solution is a hypothetical solution for which all attribute values correspond to the minimum values in the data group. TOPSIS thus provides a solution that is not only closest to hypothetically the best solution but also farthest from the hypothetical worst one.

The TOPSIS method is based on the idea that the optimal variant must have the minimum distance from the ideal solution.

The method entails defining the objective function $f: V \rightarrow R$, given by the Equation (9).

$$f(V_i) = \frac{\sum_{j=1}^{n} p_j r_{ij}}{\sum_{j=1}^{n} p_{ij}}, \ i = 1, m, \tag{9}$$

The steps of the TOPSIS method are the following:

Step 1. The normalized matrix $R = (r_{ij})$, $i = 1, \ldots, m$, $j = 1, \ldots, n$ is built;

Step 2. The weighted normalized matrix $V = (v_{ij})$, $i = 1, \ldots, m$, $j = 1, \ldots, n$ is built by Equation (10), where:

$$v_{ij} = \frac{p_j r_{ij}}{\sum_{j=1}^{n} p_j} \tag{10}$$

Step 3. The ideal solutions $A$, $B$ (Equation (11)) defined as in the Equations (12) and (13) is calculated as following:

$$\begin{aligned} A &= (a_1, a_2, \ldots, a_n) \\ B &= (b_1, b_2, \ldots, b_n) \end{aligned} \tag{11}$$

where

$$a_j = \begin{cases} max v_{ij}, if \ C_j \ max \\ \quad 1 \leq i \leq m \\ min v_{ij}, if \ C_j \ min \\ \quad 1 \leq i \leq m \end{cases} \tag{12}$$

$$b_j = \begin{cases} max \ v_{ij}, if \ C_j \ min \\ \quad 1 \leq i \leq m \\ min \ v_{ij}, if \ C_j \ max \\ \quad 1 \leq i \leq m \end{cases} \tag{13}$$

Step 4. The distances between solutions Equations (14) and (15) are calculated as follows:

$$S_i = \sqrt{\sum_{j=1}^{n} (v_{ij} - a_j)^2}, i = 1, m \tag{14}$$

$$T_i = \sqrt{\sum_{j=1}^{n} (v_{ij} - b_j)^2}, i = 1, m \tag{15}$$

Step 5. The relative proximity from the ideal solution is calculated according to Equation (16):

$$C_i = \frac{T_i}{S_i + T_i} \tag{16}$$

Step 6. The classification of the set $V$ is performed according to the descending values of $C_i$ obtained in step 5.

## 4. Case Study—Determination of Nautical Anchorage Locations in Split-Dalmatia County

The Split-Dalmatia County (Figure 3), with its seat in Split, represents the largest Croatian county by area, occupying an area of 14,045 km², of which the land part is 4572 km² (32.5%), and approximately a third is the territorial sea of Croatia with an area of 9473 km² (67.5%).

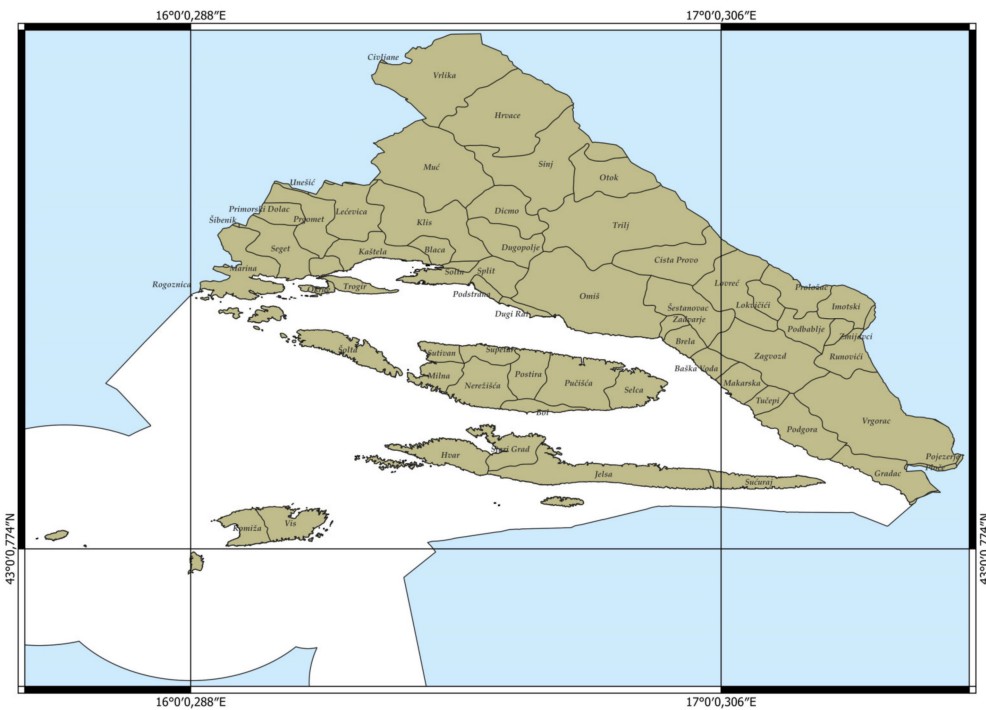

**Figure 3.** Map of Split-Dalmatia County.

A total of 455,242 inhabitants live in this county (67% in the coastal part, about 7% on the islands, and 26% in the coastal area). The anchorages in the Split-Dalmatia County are located in the central part of the eastern Adriatic coast, which is recognized as one of the most exclusive nautical destinations on the Adriatic [6].

### 4.1. Dataset

The data used as input in the research were collected in two ways. In the first part of the research, which refers to the understanding of the actual evaluations and the preferences of the characteristics of nautical anchorages from the point of view of "users", the aim is to investigate the real expectations of sailors (professionals and amateurs) regarding the conditions that must be met by nautical anchorages.

Based on the ratings of users (sailors) in the survey research, certain weighting values were assigned to the selected criteria in order to apply MCDM methods based on them, as well as on the basis of data on variants (possible locations of nautical anchorages).

The second phase of the research involves the application of MCDM methods (AHP, TOPSIS) using input data that were obtained as a result of the diligent collection, storage, analysis, and processing of many years of work and experience of the author/s, and refers to a group of 56 bays (Figure 4) with the total of their 86 nautical anchorage fields, each separately described with 20 characteristics (Figure 5).

The spatial arrangement of nautical anchorages in the area of Split-Dalmatia County is shown in the Figure 4.

On the basis of the survey research, the users (sailors) assign the greatest importance to the safety criterion and the least importance to the economic criterion, as shown in Table 3.

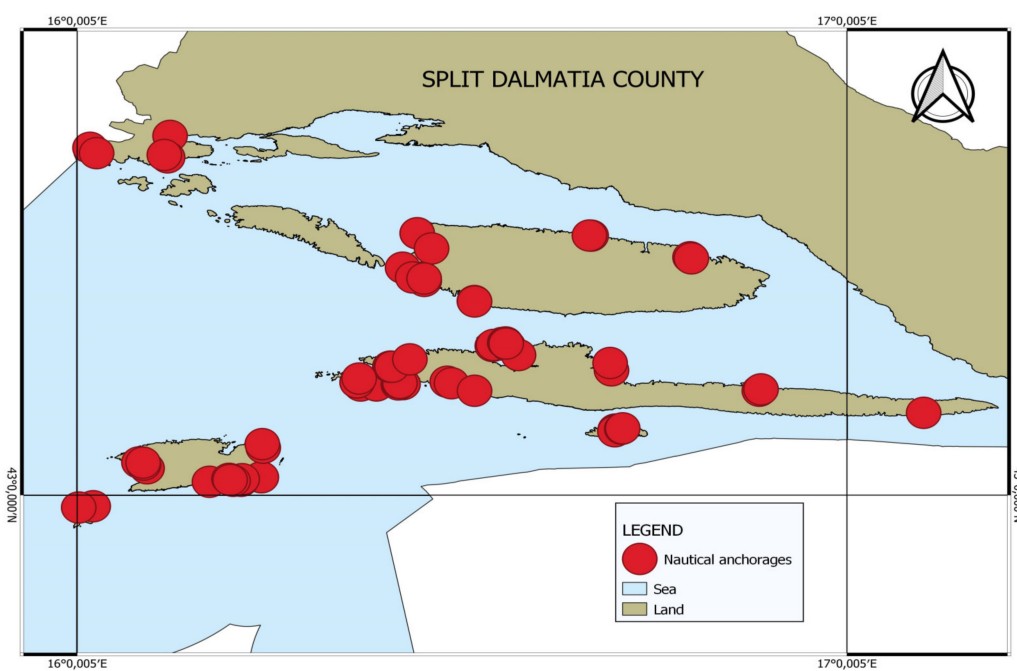

**Figure 4.** Spatial distribution of nautical anchorages in the area of Split-Dalmatia County.

| | 1 | 2 | 3 | 4 | 5 | 6 | 7 | 8 | 9 | 10 | 11 | 12 | 13 | 14 | 15 |
|---|---|---|---|---|---|---|---|---|---|---|---|---|---|---|---|
| **AHP** | 15 | 34 | 75 | 28 | 32 | 42 | 13 | 41 | 27 | 33 | 17 | 9 | 10 | 16 | 23 |
| **TOPSIS** | 15 | 34 | 42 | 75 | 41 | 37 | 38 | 77 | 79 | 78 | 36 | 28 | 43 | 33 | 23 |

**Figure 5.** Excerpt from comparative results of both applied methods of multi-criteria decision making.

**Table 3.** Survey result—sorted mean grades (from highest to lowest).

| The Name of the Criterion | Mean Grade |
|---|---|
| 1. Safety (navigation)—1.1 Underwater installations | 4.58 |
| 1. Safety (navigational)—1.2 Potential danger | 4.51 |
| 2. Hydrometeorological criteria—2.1 Protection of the bay | 4.41 |
| 5. Environmental criteria—5.3 Pollutants or waste | 4.27 |
| 3. Spatial criteria—3.2 Turning radius | 4.26 |
| 5. Environmental criteria—5.4 Local heritage | 4.15 |
| 3. Spatial criteria—3.1 Distance and depth | 4.03 |
| 1. Safety (navigation)—1.3 Manoeuvre space | 3.97 |
| 5. Environmental criteria—5.2 Disturbance of the seabed | 3.84 |
| 5. Environmental criteria—5.1 Impact on the marine environment | 3.74 |
| 4. Economic criteria—4.2 Public anchorage | 3.30 |
| 2. Hydrometeorological criteria—2.2 Current and tide | 3.26 |
| 4. Economic criteria—4.5 Profitability | 3.19 |
| 4. Economic criteria—4.1 Port area | 2.82 |
| 4. Economic criteria—4.3 Access to land | 2.81 |
| 4. Economic criteria—4.4 Traffic and other infrastructure | 2.55 |

For this reason, in the second part of the research, the security elements (available data that were treated as criteria) were assigned the maximum weight values when applying the multi-criteria analysis methods as follows.

The data that have been collected are shown in the Table 3, demonstrating that out of all the important criteria and factors significant for nautical anchorages, users (sailors) emphasize safety factors, namely the following: distance from existing underwater installations from anchorages; protection of the anchorage from wind and waves; the size of the

manoeuvring surface, etc., so that in the second phase, when applying the multi-criteria analysis methods, the safety criteria (navigation—protection of the anchorage; the surface of the anchorage field; the surface of the bay; the distance from the coast) will be assigned the highest weight values, as well as the environmental criteria.

The data that have been collected (Table 4) on the locations of nautical anchorages in the area of Split-Dalmatia County are as follows:

- Serial number of the location (No.);
- Location name (Name);
- Island (Island);
- The name of the field in the bay (Field);
- Field surface (Surface F);
- Surface of the bay (Surface B);
- The percentage of the field surface in the bay area (Percentage);
- Protection of the bay (1. Protected; 5: Partially protected; 9: Not protected) (Protection);
- Distance from the coast (Distance);
- Number of anchorage fields in the bay (Number F);
- Existence of maritime traffic (Traffic);
- Existence of an official anchorage (Anchorage);
- Existence of underwater cables and pipelines (Cables);
- Risk of collision (Danger);
- Depth (Depth);
- Tide level and existence of sea currents (Tide);
- Proximity to public ports (Proximity P);
- Proximity to existing berths (Existing B);
- Elements of the environment (Environmental network Natura 2000) (Environment);
- Harm from anchoring a vessel to the seabed (Harmfulness);
- Archaeological sites (Site).

**Table 4.** Excerpt from collected data on the first eight locations of nautical anchorages.

| No | 1 | 2 | 3 | 4 | 5 | 6 | 7 | 8 |
|---|---|---|---|---|---|---|---|---|
| Name | MILNA Lucice | MILNA Lucice | MILNA Lucice | MILNA Mali bok | MILNA Osibova uvala | MILNA Uvala Slavinjina | NEREZISCA Uvala Blaca | NEREZISCA Uvala Blaca |
| island | BRAC | BRAC | BRAC | BRAC | BRAC | BRAC | BRAC | BRAC |
| field | A | B | C | A | A | A | A | B |
| surface F | 2059.39 | 17,781.05 | 6744.95 | 3416.15 | 5176.56 | 7756.37 | 5990.29 | 2400 |
| surface B | 723,767.14 | 723,767.135 | 723,767.135 | 11,800.58 | 227,733.312 | 64,415.177 | 71,424.38 | 71,424.378 |
| percentage | 0.2845377 | 2.45673631 | 0.93192267 | 28.94899 | 2.27307984 | 12.0412151 | 8.386898 | 3.3601973 |
| openness | 5 | 5 | 5 | 9 | 5 | 5 | 5 | 5 |
| distance | 3.9 | 24.3 | 24 | 12.4 | 2.7 | 15.2 | 15.8 | 0 |
| numberoF | 3 | 3 | 3 | 1 | 1 | 1 | 2 | 2 |
| traffic | 1 | 1 | 1 | 1 | 1 | 1 | 1 | 1 |
| anchorage | 1 | 1 | 1 | 1 | 1 | 1 | 1 | 1 |
| underwaterI | 1 | 1 | 1 | 1 | 1 | 1 | 1 | 1 |
| danger | 3 | 3 | 3 | 3 | 3 | 3 | 3 | 3 |
| depth | 5 | 5 | 5 | 5 | 5 | 5 | 5 | 5 |
| tide | 1 | 1 | 1 | 1 | 1 | 1 | 1 | 1 |
| proximityoP | 1 | 1 | 1 | 1 | 1 | 1 | 1 | 1 |
| existingA | 1 | 1 | 1 | 1 | 1 | 1 | 1 | 1 |
| environment | 5 | 5 | 5 | 5 | 5 | 5 | 5 | 5 |
| harmfulness | 5 | 5 | 5 | 5 | 5 | 5 | 5 | 5 |
| site | 1 | 1 | 1 | 1 | 1 | 1 | 1 | 1 |

*4.2. Description of the Criteria and Initial Settings for the Application of MCDM Methods*

The description of the designation and name of the criteria, the unit of measure, and the range of each criterion are defined in Table 5.

**Table 5.** Label, criterion name, measure unit, and range of input data.

| Label | Criterion Name | Measure Unit | Range | |
|---|---|---|---|---|
| | | | Min | Max |
| C1 | Field surface (Surface F) | m$^2$ | 900 | 76,654.4 |
| C2 | Area of the bay (Surface B) | m$^2$ | 11,800.6 | 723,767 |
| C3 | The percentage of the field area in the bay area (percentage) | % | 0.28454 | 39.578 |
| C4 | Protection of the bay (Protection) | Whole number: 1—Protected bay; 5—Partially protected; 9—Non protected | 1 | 9 |
| C5 | Distance from the coast (Distance); | m | 0 | 81.2 |
| C6 | Number of anchorage fields in the bay (Number F) | Whole number | 1 | 4 |
| C7 | Existence of maritime traffic (Traffic) | Whole number: If the proximity of the main traffic routes is less than 500 m: 1—No; 5—Yes | 1 | 5 |
| C8 | Existence of an official anchorage (Anchorage) | Whole number: If it is in the area of official anchorages: 1—No; 5—Yes | 1 | 5 |
| C9 | Existence of underwater cables and pipelines (Cables) | Whole number: The proximity of cables and pipelines is less than 500 m: 1—No; 5—Yes | 1 | 5 |
| C10 | Risk of collision (Danger) | Whole number: 1—negligibly small; 2—small; 3—mean; 4—big; 5—very big | 1 | 5 |
| C11 | Depth (Depth) | Whole number: 1—Satisfactory; 5—Unsatisfactory | 1 | 5 |
| C12 | Tide level and existence of sea currents (tide) | Whole number: 1—small; 3—mean; 5—big | 1 | 5 |
| C13 | Proximity to public ports (Proximity P) | Whole number: 1—No; 5—Yes | 1 | 5 |
| C14 | Proximity to existing berths (Existing B) | Whole number: 1—No; 5—Yes | 1 | 4 |
| C15 | Elements of the environment (Environmental network Natura 2000) (environment) | Whole number: 1—No; 5—Yes | 1 | 5 |
| C16 | Harm from anchoring a vessel to the seabed (harmfulness) | Whole number: 1—No; 5—Yes | 1 | 5 |
| C17 | Archaeological sites (sites) | Whole number; 1—No; 5—Yes | 1 | 5 |

Based on the Saaty scale, the relationship between the criteria for the application of the AHP method (Table 6).

The complicated part of the application of the AHP method is the creation of a consistent decision matrix and the establishment of a consistent correlation between the criteria which is, due to the number of criteria (ten), very difficult to implement.

Based on the relationship matrix between the criteria in the AHP method, the value of the maximum eigenvalue ($\lambda$max = 11.2671), the consistency ratio (CI = 0.14079), and the consistency index, which is less than 10% (CR = 0.09449), a correct relationship between the criteria is considered established. The consistency index (CR), being less than 10%, also represents a good ratio established between the criteria.

The names of the criteria, the target vector, and the vector of weight coefficients in the TOPSIS method (Table 7).

**Table 6.** Relationship between criteria according to the Saaty scale (AHP).

| Criterion Name | | Surface F | Surface B | Percentage | Protection | Distance | Number F | Number F | Existing B | Environment | Harmfulness |
|---|---|---|---|---|---|---|---|---|---|---|---|
| Criterion name | Designation | C1 | C2 | C3 | C4 | C5 | C6 | C7 | C8 | C9 | C10 |
| Surface F | C1 | 1 | 1 | 1 | 1 | 2 | 6 | 1/2 | 2 | 2 | 5 |
| Surface B | C2 | 1 | 1 | 1 | 1 | 1 | 5 | 1/2 | 2 | 2 | 4 |
| Percentage | C3 | 1 | 1 | 1 | 4 | 2 | 7 | 5 | 3 | 3 | 4 |
| Protection | C4 | 1 | 1 | 1/4 | 1 | 1 | 7 | 2 | 5 | 5 | 5 |
| Distance | C5 | 1/2 | 1 | 1/2 | 1 | 1 | 4 | 3 | 2 | 2 | 3 |
| Number F | C6 | 1/6 | 1/5 | 1/7 | 1/7 | 1/4 | 1 | 1/9 | 1/2 | 1/2 | 1 |
| Number F | C7 | 2 | 2 | 1/5 | 1/2 | 1/3 | 9 | 1 | 4 | 4 | 9 |
| Existing B | C8 | 1/2 | 1/2 | 1/3 | 1/5 | 1/2 | 2 | 1/4 | 1 | 1 | 2 |
| Environment | C9 | 1/2 | 1/2 | 1/3 | 1/5 | 1/2 | 2 | 1/4 | 1 | 1 | 3 |
| Harmfulness | C10 | 1/5 | 1/4 | 1/4 | 1/5 | 1/3 | 1 | 1/9 | 1/2 | 1/3 | 1 |

**Table 7.** Elements of the goal vector and their weight values when applying TOPSIS method.

| TOPSIS | Criterion Name | Goal Vector | Vector of Weighting Coefficients |
|---|---|---|---|
| Criterion/(name of criteria in the tables) | Label | pi | wi |
| Field surface (Surface F) | C1 | max | 5 |
| Area of the bay (Surface B) | C2 | max | 4 |
| The percentage of the field area in the bay area (Percentage) | C3 | max | 9 |
| Protection of the bay (1. Protected; 5: Partially protected; 9: Not protected) (Protection) | C4 | min | 13 |
| Distance from the coast (Distance) | C5 | max | 3.5 |
| Number of anchorage fields in the bay (Number F) | C6 | min | 1 |
| Existence of maritime traffic (Traffic) | C7 | min | 1 |
| Existence of an official anchorage (Anchorage) | C8 | min | 1 |
| Existence of underwater cables and pipelines (Cables) | C9 | min | 1 |
| Risk of collision (Danger) | C10 | min | 1 |
| Depth (Depth); | C11 | max | 1 |
| Tide level and existence of sea currents (Tide) | C12 | min | 9 |
| Proximity to public ports (Proximity P) | C13 | min | 1 |
| Proximity to existing berths (Existing B) | C14 | min | 1 |
| Elements of the environment (Environmental network Natura 2000) (Environment) | C15 | min | 2 |
| Harm from anchoring a vessel to the seabed (Harmfulness) | C16 | min | 2 |
| Archaeological sites (Sites) | C17 | min | 1 |

## 5. Result—Validation, Testing, and Comparison Analysis

Obtaining results using two methods of multi-criteria analysis, AHP and TOPSIS, implies [31] the following:

- selection of possible decision options;
- selection of evaluation criteria;
- obtaining performance measures and their fulfilment;
- transformation of data into proportional units (depending on the type of multi-criteria technique method that is applied), which mostly requires entries of decision-makers' preferences;

- determination of criteria and their weighting values, which also largely depends on the preferences of decision-makers;
- ranking or scoring options;
- implementation of sensitivity analysis (weight, performance measures, technique);
- making selection decisions.

In this paper, numerous criteria are analysed in order to determine the most favorable locations of anchorages meeting the conditions prescribed by the recommendations [2,3], while also meeting the expectations of future users, spatial planners, potential investors, and concessionaires who would operate in these areas, as well as entities striving to preserve and protect marine and underwater animal life and the environment, as well as prevent its degradation and pollution. However, since the given recommendations are not precisely defined for the establishment of nautical anchorages, in the procedures for determining the location of nautical anchorages, general and specific criteria are used to satisfy them.

The achieved results indicate that implementation of the methods of multi-criteria analysis enables the selection of the best locations of nautical anchorages in the area of Split-Dalmatia County.

The available criteria (17 in total) for each of the 86 locations represent a decision-making matrix in the implementation of the multi-criteria analysis method. In the AHP method, the relationship between 10 criteria was defined, while in the TOPSIS, all 17 criteria were used, which were assigned goals and weight values.

The list of the sequence of the best locations (per the ordinal numbers of the locations from the input dataset) from the best to the worst (15 of the initial 86) are shown in Figure 5. The same figure indicates the comparative results of two comparisons, shown in order to exhibit the match.

The data (Figure 5) show that, although the order of the 15 best locations obtained by multi-criteria analysis methods is not the same, they differ in a maximum 7 locations.

If the results of the order of the first 15 locations are compared, it is observed that they do not match in a maximum of 7 locations.

The data in Figure 5 show the ordinal numbers of the locations that are in the first 15 best locations obtained by the AHP and TOPSIS method. Shaded (pink) are the ordinal numbers of locations that are in the list of the best 15 obtained both by AHP and based on the TOPSIS method, while those that are not shaded are not recognized as the best by both the AHP and the TOPSIS method. So, locations numbered: 32, 13, 27, 17, 9, 10, 37, 38, 77, 79, 78, 36, and 43 do not appear as the best in both methods (AHP and TOPSIS). Locations marked with numbers 32, 13, 27, 17, 9, 10, and 16 belong to the list of the best 15 locations according to the AHP method, but are not recognized in the list of the best obtained by the TOPSIS method. At the same time, the locations marked with numbers 37, 38, 77, 79, 78, 36, and 43 belong to the list of the best obtained by the TOPSIS method, but AHP method does not recognize them in the list of the best 15.

The sensitivity analysis of the results obtained by the AHP method performed based on the relationship matrix between the criteria. The value of the maximum eigenvalue ($\lambda$max = 11.2671), the consistency ratio (CI = 0.14079), while the consistency index is CR = 0.09449 and is less than 10%. The Consistency Index (CR) which, being less than 10%, represents a good ratio established between the criteria.

The reason for this can be found in the difference in the initial settings, the difference in the number, the relationship between the criteria, and the difference in the calculation of the multi-criteria analysis method that are analysed and considered.

Therefore, considering that both applied methods of multi-criteria analysis mostly gave the same or similar results, the research with the proposed methods and the obtained solution represent a strong and effective decision-support tool in the further planning and decision-making process.

## 6. Discussion

In order to check the robustness, reliability, stability, and accuracy of the obtained solutions, internal and external validation of the results obtained by the TOPSIS method are used.

Internal validation is meant to check the stability of the results by verifying their consistency by comparing the obtained results with data that partially or slightly changes in order to observe the changes that occur. External validation was obtained on the basis of survey research, i.e., by applying a different (alternative) research technique.

As the data on the test (additional/validated) locations were obtained on the basis of survey research (according to other research techniques), and not on the basis of multi-criteria analysis methods, the results are compared by partially changing or adding new data (about five new locations).

As part of the survey, the respondents had the opportunity to (graphically) propose the locations of nautical anchorages (at the site within a group of 86 locations), and to propose both the form and the size of the field of nautical anchorages. The data were cleaned (valueless and unusable ones being rejected), and based on such arranged and organized data, the values of five test locations, i.e., fields, their size and distance from the coast, were generated. As a result of the data obtained in this way, the area of the fields, the share of the percentage of the area of the field in the area of the bay, as well as the distance from the coast, were calculated.

For each of the five proposed test locations (suggested by the surveyed respondents), the input data are identical to the data of the five existing locations from the group of 86 sites, except for the following: 1. field surface; 2. percentage share of the field surface in the bay; and 3. distance from the coast.

The five test locations (Table 8) are marked as follows: 4T, 11T, 42T, 46T, and 74T. Data for the locations with which the order of the obtained results and the values of the elements that change are compared for the locations marked with numbers: 4 and 4T; 11 and 11T; 42 and 42T; 46 and 46T; and 74 and 74T.

**Table 8.** Location data being validated and location data they are compared with.

| No. | 4 | 4T | 11 | 11T | 42 | 42T | 46 | 46T | 74 | 74T |
|---|---|---|---|---|---|---|---|---|---|---|
| Surface F | 3416.2 | 3278.2 | 15,056.1 | 16,588.6 | 33,300.6 | 46,613.6 | 11,217.8 | 12,947.1 | 16,066.3 | 14,430.2 |
| Percentage | 29.0 | 27.8 | 15.0 | 16.5 | 19.3 | 27.0 | 4.8 | 5.6 | 21.4 | 19.2 |
| Distance | 12.4 | 5.6 | 7.4 | 30.2 | 15.6 | 12.6 | 7.4 | 29.8 | 23.2 | 22.5 |

Considering that it would be utterly impractical and highly unnecessary to validate the results through both MCDM method used in the empirical research (due to their similarity), the data are validated/tested using the TOPSIS method and include a dataset of 91 locations 5 new and 86 initial locations).

The validation results are presented in Table 9.

**Table 9.** Ranking and score of testing dataset.

| Rank | Score | No. |
|---|---|---|
| 2 ˆ↑ | 0.677367199 | 42T |
| 4 | 0.608256708 | 42 |
| 18 | 0.527340126 | 74 |
| 23 ˇ↓ | 0.512747238 | 74T |
| 34 | 0.479203486 | 4 |
| 40 ˇ↓ | 0.471523883 | 4T |
| 47 ˆ↑ | 0.456666886 | 11T |
| 61 | 0.438131969 | 11 |
| 89 ˆ↑ | 0.359936881 | 46T |
| 90 | 0.347016121 | 46 |

The validation results indicate very small sequence changes. The location was marked with the number 4T, which the survey respondents proposed to have a surface of 3278.2 m$^2$, with a share of 27.78% in the surface of the bay, a surface of 11,800,584 m$^2$, and a distance from the coast of 5.6 m, instead of the area of 3416.15 m$^2$ that has location 4, with a share of 28.9% in the surface of the bay and the distance from the coast of 12.4 m that location 4 had, occupies the 40th position (row 7 of Table 9.) instead of the 34th (row 6 of Table 9.) that was occupied by the location marked with ordinal number 4. Therefore, given that location 4 occupied 34th place, and the location labelled 4T ranks 40th on the list, no significant change has occurred. However, neither the location 4T nor the location marked with serial number 4 belong to the group of the 15 best locations for nautical anchorages. The same happens with the locations marked with the serial numbers 74 and 74T, as well as the locations marked with the serial numbers 11 and 11T and 46 and 46T.

The locations marked with serial numbers 42 and 42T are in the fourth and second positions, respectively, confirming that location 42 (as well as 42T) is an excellent choice for the location of a nautical anchorage. Both are on the list of the top 15 nautical anchorage locations, with the location marked 42T now taking the second position and the location marked 42 the fourth.

This means that although the results achieved at both locations (42 and 42T) are equally good, if it had to be chosen between the data that covered the locations marked 42T and 42, it would be better to decide upon the field surface and distance that were registered at the location marked 42T.

Based on the application of both internal and external validation of the obtained results, we can conclude that the obtained results are very stable and consistent, and that significant changes in the order of the best 15 locations would not change due to insignificant/or minor changes in the input data.

## 7. Conclusions

The purpose and goal of this paper was to demonstrate the usability of MCDM methods in the process of optimizing the location of nautical anchorages in the Split-Dalmatia County. The MCDM methods include the application of AHP and TOPSIS, based on which the 15 best from the group of 86 possible variants, i.e., locations, are selected in the selection process. The evaluation and assignment of weight values of the criteria were performed based on the ratings given by the respondents (in the first phase) to the most important elements of nautical anchorages. The data were collected through a questionnaire that was created, distributed, and filled in by 74 users (amateur and professional sailors). Thus, the most important criteria from the user's point of view were assigned the highest weight values when applying MCDM methods (second phase), and they mainly relate to the safety of the nautical anchorage. The results showed that both applied MCDM methods, with different initial settings specific to each method (by determining the relationship between the criteria, their weight values, and the objective function of each criterion), gave very similar solutions. The obtained lists of the best MCDM solutions (the most differ in 6 and the least differ in 3) confirm the effectiveness of the MCDM method in selecting the best nautical anchorage locations in Split-Dalmatia County. The obtained consistency ratio among the criteria when applying the multi-criteria AHP method was 9.449231, and is considered acceptable given that they are less than 10% and reflect a good assessment of the criteria and their relationships for the considered case study example. The list of the best 15 locations of nautical anchorages can be used for various deeper and broader research by professional sailors and amateurs, spatial planners, future concessionaires, county offices, scientific and professional staff, and all interested parties, as well as scientific and educational institutions.

As the work takes into account the selection of the best locations of nautical anchorages for smaller vessels and yachts, it is recommended to apply multi-criteria analysis methods and select adequate criteria for larger vessels and/or merchant ships in the future.

Further research may refer to the expansion of multi-criteria decision-making methods by including a larger group of data (other and more numerous locations and criteria, such as vessel size, bottom type, etc.) when selecting the best locations for nautical anchorages for a wider area than the Split-Dalmatia County, for example, the entire area of the Adriatic Sea and beyond. The outcome of this could be a contribution to documenting and enriching the relevant literature in the field of application of multi-criteria decision-making methods in solving more complex problems, and not only in the fields of seafaring and maritime safety, spatial planning, shipping, nautical tourism, and maritime economy.

**Author Contributions:** Conceptualization, D.P. and Z.L.; methodology, Z.L.; software, D.P.; validation, D.P.; formal analysis, D.P. and Z.L.; investigation, D.P.; resources, D.P.; writing—original draft preparation, D.P.; writing—review and editing, D.P. and Z.L.; visualization, D.P.; supervision, Z.L. All authors have read and agreed to the published version of the manuscript.

**Funding:** This research received no external funding.

**Institutional Review Board Statement:** Not applicable.

**Informed Consent Statement:** Not applicable.

**Data Availability Statement:** Not applicable.

**Conflicts of Interest:** The authors declare no conflict of interest.

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
