# Peer review of "Multi-Criteria Decision Analysis for Nautical Anchorage Selection"

_jmse, doi:10.3390/jmse11040728_

Round 1

Reviewer 1 Report

Reviewer Recommendation and Comments for manuscript jmse-2303375-peer-review-v1.pdf with the title: “Multi Criteria Decision Analysis for Nautical Anchorage Selection”, with follow authors Danijel Pušić and Zvonimir Lušić.

The goal of this manuscript is was to demonstrate the usability of MCDM methods in the process of optimizing the location of nautical anchorages in the Split-Dalmatia County.

This is a relevant research which analyses dealing with Multi Criteria Decision Analysis for Nautical Anchorage in the specific area. The text is clearly written. The structure, content, and concept of the research work as well as the achievements, correspond to the new unpublished scientific paper. The English is fine and the paper is clearly written.

The abstract is clear and in the proper way present the scope, materials, methods, achievements, and future research directions. The introduction is fine. Materials and Methods consist Methodology and research plan research background and recommendation. Using methods of multi-criteria analyses well clear presented and described. Case Study, results and discussion are fine.

Tables, Figures, and math equations are properly presented.

In the conclusion, authors specify purpose and goal of the research and explains the achievements of this original scientific research. Future research directions are listed.

 NOTE: I believe that citing literary sources is an error in the automatic processing of citing references and I believe that it will be corrected before the final publication of the work.

Author Response

Dear Reviewer,

We would like to thank you for taking the time to read this paper, as well as for your correct and comprehensive suggestions. All this contributes to this paper meeting the high standards and conditions defined by you and the publisher. We must note that we strictly followed the instructions on writing the paper, and tried to explain the methods and steps, procedures and data used in a simple but high-quality way, in order to ensure the selection of the best locations of nautical anchorages in the area of Split-Dalmatia County.

Point 1: I believe that citing literary sources is an error in the automatic processing of citing references and I believe that it will be corrected before the final publication of the work.

Response 1: Certainly, we will ensure that all citations are thoroughly checked and corrected in the revised manuscript.

Thank you for bringing this to our attention, and please let us know if you notice any other errors or inconsistencies.

Regards,

Danijel Pušić

Reviewer 2 Report

Article: Multi Criteria Decision Analysis for Nautical Anchorage Selection

Anchoring, as one of the routines and frequently performed operations in the maritime industry, has many negative impacts, so it is necessary to pay much more attention to it and carry out detailed analyses and research related to its improvement. This paper's topic is highly current, and its concrete results are significant for future research. The article gives a detailed description of the criteria based on which it is possible to choose the best position for anchoring the ship, taking into account many factors.

There are a few suggestions for the authors:

1.        The abbreviations appearing in the text should be explained/given in full when appearing for the first time.

2. It is necessary to check the cited literature and indicate the reference numbers in the text using the same type (style) of numbers that are listed in the literature review.

3. Section 2.1's subsection 'Validation, testing, comparison and result analysis' should be highlighted.

4.        In Table 1, it is necessary to check the elements of the matrix.

5.        It is necessary to check the matrix elements in equation (2).

6.        Figure 5 is unclear.

7.        Section 5. 'Results' needs to be more comprehensive and with more specific details.

8.        It is necessary to give a more detailed explanation of the results shown in Figure 6. Figure 6 is a Table, so it should be marked that way.

9.        In future research, it is recommended to analyze in detail the type and the vessel's size (dimensions) as a criterion for choosing an anchorage.

10.      English has to be checked and corrected. A certain number of sentences lose their meaning due to the lack of commas and the irregular order of words in the sentence.

Author Response

Dear Reviewer,

Thank you for taking the time to review our paper and for providing us with your valuable feedback. Your insightful suggestions have helped us to improve the quality and standards of our work.

We would like to emphasize that we followed the instructions for writing this paper carefully and made every effort to present our methods, procedures, and data in a clear and concise manner. Our goal was to identify the best locations for nautical anchorages in the area of Split-Dalmatia County, and we are confident that our approach and findings will be of value to readers.

We provide answers to your questions and suggestions below.

Point 1: The abbreviations appearing in the text should be explained/given in full when appearing for the first time.
Response 1: Corrected.

Point 2: It is necessary to check the cited literature and indicate the reference numbers in the text using the same type (style) of numbers that are listed in the literature review.
Response 2: The numbering of the literature follows the instructions of the publisher. Square brackets in the text, ordinal Arabic numbers at the end. Nevertheless, at the end, the list of used literature was made in such a way that both in the text and in the list of references, the cited sources correspond to the number in the list of references using square brackets.

Point 3: Section 2.1's subsection 'Validation, testing, comparison and result analysis' should be highlighted.
Response 3: Validation, testing, comparison and result analysis are found in the section related to results (Chapter 5) - 5. Result – Validation, testing and comparison analysis.

Point 4: In Table 1, it is necessary to check the elements of the matrix.
Response 4: Corrected - Added "=" missing.

Point 5: It is necessary to check the matrix elements in equation (2).
Response 5: Corrected - Written in small letters, some letters were capitalized, which was a mistake.

Point 6: Figure 5 is unclear.  
Response 6: Considering that the images and tables should be located on almost half the width of the page, the table had to be narrowed in the available space. The number of columns is large (20), and the number of rows is 86. That is why it is displayed in such a way that only a small segment of the table is visible. The image is placed in the middle of the page.

Point 6: Section 5. 'Results' needs to be more comprehensive and with more specific details.
Response 6: We added text from line 594-616 and 635-651.

Point 7: It is necessary to give a more detailed explanation of the results shown in Figure 6. Figure 6 is a Table, so it should be marked that way.
Response 7: In the results, a text has been added that describes in more detail all the steps that explain the procedures for selecting criteria, evaluating weight values, relationships between criteria, etc. See chapter 5. Result – Validation, testing and comparison analysis.

Point 8: In future research, it is recommended to analyze in detail the type and the vessel's size (dimensions) as a criterion for choosing an anchorage.
Response 8: As the work takes into account the selection of the best locations of nautical anchorages for smaller vessels and yachts, in the future it is recommended to apply multi-criteria analysis methods and select adequate criteria for larger vessels and/or merchant ships, including their dimensions. Regarding dimensions and future research few words added to Conclusions.

Point 9: English has to be checked and corrected. A certain number of sentences lose their meaning due to the lack of commas and the irregular order of words in the sentence.
Response 9: In this part, the authors asked for the help of an English language teacher, who carefully read the paper and corrected the mistakes.

We would like to inform you that we have taken all of your suggestions and corrections into consideration and will ensure that they are included in the revised manuscript.

Thank you once again for your time and effort in reviewing our paper.

Regards,

Danijel Pušić

Reviewer 3 Report

- The authors worked out the case by applying a new approach to solving the problem to determine the most favorable/optimal locations for nautical anchorages by applying multi-criteria analysis.

- The paper has an excellent structure with a clear explanation.

- The steps in creating a paper could be better explained.

- The developed methodology is given in detail.

- Good problem definition.

The paper has great potential and can be accepted after the following MAJOR corrections:

1. It is necessary to clarify more precisely how five groups of elements (and a total of 18 sub-elements) were reached. One gets the impression that the authors independently proposed and then offered the construction of the elements in the survey questionnaire! (line 122)

2. It is necessary to list the sources that helped to define the parameters for the multicriteria analysis (or the way in which the parameters were defined). (Line 144)

3. Technical processing: pictures 1 (order the row numbers); arrange tables in text format (Table 4, 5 and 6)

4. Harmonize the way of indicating the reference in the text (whether Arabic or Roman numerals are used, or whether the name of the source is also indicated (Line 147) and in the list of references (Arabic numerals)

5. How was the sensitivity analysis of the results obtained by the AHP method performed?

6. The list of bibliographic references seems rather short. I believe that authors should increase the number of cited references with which they can report and compare their research findings. Perhaps papers with different problems could provide a clearer picture for solving the above problem. Papers that might help are:

6.1. Alosta, A., Elmansuri, O., & Badi, I. (2021). Resolving a location selection problem by means of an integrated AHP-RAFSI approach. Reports in Mechanical Engineering, 2(1), 135-142. https://doi.org/10.31181/rme200102135a

6.2.Badi, I., & Abdulshahed, A. (2021). Sustainability performance measurement for Libyan Iron and Steel Company using Rough AHP . Journal of Decision Analytics and Intelligent Computing, 1(1), 22–34. https://doi.org/10.31181/jdaic1001202222b

Author Response

Dear Reviewer,

We would like to thank you for taking the time to read this paper, as well as for your correct and comprehensive suggestions. All this contributes to this paper meeting the high standards and conditions defined by you and the publisher. We must note that we strictly followed the instructions on writing the paper, and tried to explain the methods and steps, procedures and data used in a simple but high-quality way, in order to ensure the selection of the best locations of nautical anchorages in the area of Split-Dalmatia County.

We provide answers to your questions and suggestions below.

Point 1: It is necessary to clarify more precisely how five groups of elements (and a total of 18 sub-elements) were reached. One gets the impression that the authors independently proposed and then offered the construction of the elements in the survey questionnaire! (line 122)

Response 1:
There are a number of factors that need to be considered when identifying suitable sites for anchoring vessels and will often require consultation with a wide range of stakeholders. The most important guidelines and recommendations on factors to consider  are defined by: Queensland Government, Anchorage Area Design and Management Guideline, Maritime Safety Queensland, 2019. p. 3. and PIANC, Report n° 121 - 2014, PIANC Secrétariat Général, Bruxelles, 2014. (https://www.pianc.org/technicalreportsbrowseall.php) lines 169-183. The most important elements on the basis of which the criteria were defined and grouped into five groups as shown in Figure 1 are: Anchorage places; The depth of the sea; Types of anchorage bottoms; Meteorological and hydrographic factors; Layout and infrastructure of the anchorage; Size and arrangement of anchorages; Management of the anchorage; Anchorage assignment rules; Rules for using anchorages and communication; Discharge of pollutants or waste from vessels; Activities of vessels at anchor; Reducing the risk of anchoring due to a collision; Dangers of vessel impact or grounding; Controls and supervision of anchorages; Port services; Lost anchors; Emergency situations; Considerations of environmental factors; Environmental impact assessment; Impacts on the seabed; Management of pollutant emissions or waste; Aesthetic factors; Marine pests; Conservation dependent species; Local heritage values and others as defined by The World Association for Waterborne Transport Infrastructure (PIANC). Paper corrected (line 149-160) accordingly.

Point 2: It is necessary to list the sources that helped to define the parameters for the multicriteria analysi (or the way in which the parameters were defined). (Line 144)

Response 2: The most important guidelines and recommendations on factors to consider  are defined by: 1. Queensland Government, Anchorage Area Design and Management Guideline, Maritime Safety Queensland, 2019. p. 3. and 2. PIANC, Report n° 121 - 2014, PIANC Secrétariat Général, Bruxelles, 2014. (https://www.pianc.org/publications/marcom/harbour-approach-channels-design-guidelines). This should be visible from the citation in the text, also in answer to question 1.

Point 3: Technical processing: pictures 1 (order the row numbers); arrange tables in text format (Table 4, 5 and 6).

Response 3:  Corrected.

Point 4: Harmonize the way of indicating the reference in the text (whether Arabic or Roman numerals are used, or whether the name of the source is also indicated (Line 147) and in the list of references (Arabic numerals)

Response 4: The numbering of the literature follows the instructions of the publisher. Square brackets in the text, ordinal Arabic numbers at the end. Nevertheless, at the end, the list of used literature was made in such a way that both in the text and in the list of references, the cited sources correspond to the number in the list of references using square brackets.

Point 5: How was the sensitivity analysis of the results obtained by the AHP method performed?

Response 5: The sensitivity analysis of the results obtained by the AHP method performed based on the relationship matrix between the criteria. The value of the maximum eigenvalue (Lamdamax =11.2671), the Consistency Ratio (CI=0.14079), while the Consistency Index which is CR=0.09449 and it is less than 10%. The Consistency Index (CR) which, being less than 10%, represents a good ratio established between the criteria. (line 584-593 and 798-803 updated).

Point 6: The list of bibliographic references seems rather short. I believe that authors should increase the number of cited references with which they can report and compare their research findings. Perhaps papers with different problems could provide a clearer picture for solving the above problem. Papers that might help are:

Response 6: We added 5 more sources, as well as the sources you suggested and cite them in the background.

6.1. Alosta, A., Elmansuri, O., Badi, I. (2021). Resolving a location selection problem by means of an integrated AHP-RAFSI approach. Reports in Mechanical Engineering, 2(1), 135-142. https://doi.org/10.31181/rme200102135a

6.2. Badi, I., Abdulshahed, A., Sustainability performance measurement for Libyan Iron and Steel Company using Rough AHP, ata, Libya, Journal of Decision Analytics and Intelligent Computing Vol. 1 No. 1, 2021., DOI: https://doi.org/10.31181/jdaic1001202222b.

We would like to inform you that we have taken all of your suggestions and corrections into consideration and will ensure that they are included in the revised manuscript.

Thank you once again for your time and effort in reviewing our paper.

Regards,

Danijel Pušić

Round 2

Reviewer 3 Report

Sincere congratulations on the fast and professional attitude towards the improvement of the work.